# Analyzing the Contribution of Commonsense Knowledge Sources for Why-Question Answering

**Yash Kumar Lal**[*]
Stony Brook University

**Horace Liu**
Stony Brook University

**Niket Tandon**
Allen Institute for AI

**Nathanael Chambers**
US Naval Academy

**Raymond Mooney**
University of Texas, Austin

**Niranjan Balasubramanian**
Stony Brook University

## Abstract

Answering questions about why events happen in narratives requires commonsense knowledge that is external to the narrative. What aspects of this knowledge is accessible to large models? What aspects can be made accessible via external commonsense resources? We study these in the context of answering Why questions in the TellMeWhy dataset using COMET as a source of relevant commonsense relations. We analyze the relative improvements over a base T5 model when (a) increasing the model size, (b) injecting knowledge from COMET as part of the task input, and (c) asking the model to generate COMET relation type as an explanation in addition to its answer. Results show that the larger model, as expected, yields substantial improvements over the base. Interestingly, we find that the question specific COMET relations can provide substantial improvements for both base and large models, with additional possible gains when asking the model to also generate COMET relation type. So, we augment a large model with noisy hints from COMET and find that this improves performance on the *TellMeWhy* task. We also develop a simple ontology of knowledge types and analyze the relative coverage of the different models on these categories. Together, these findings suggest potential for methods that can automatically select and inject commonsense from relevant sources.

## 1 Introduction

Humans reason about events in narratives by making inferences about why those events happen. The recently introduced *TellMeWhy* dataset tests for this capability by posing *why* questions over events in simple narratives (Lal et al., 2021). Answering these often requires commonsense knowledge (CSK) that is not explicitly stated as part of the narratives. Indeed, QA models built over standard *base* sized language models fare poorly, especially on questions where the answer is not directly available in the narrative.

There are two broad avenues for incorporating the necessary commonsense knowledge for this task. One is to look at even larger language models (e.g. T5-11B (Raffel et al., 2020)) and another is to leverage external knowledge resources. The former can be seen as an implicit approach where we tap knowledge that is acquired via language modeling and general QA task pretraining. The latter is an explicit approach where we inject knowledge from a resource as part of the context. Specifically, we ask three follow-up questions that can inform further research along these avenues: **(1)** What aspects of commonsense knowledge are already accessible to larger language models? **(2)** What aspects can be made accessible by injecting information from relevant knowledge sources? **(3)** What kinds of knowledge remains inaccessible?

For the TellMeWhy task, we explore the utility of COMET[1] (Bosselut et al., 2019; Hwang et al., 2021) as a knowledge source. COMET is a transformer-based model that can generate commonsense inferences about events that it has learned from ATOMIC (Sap et al., 2019; Hwang et al., 2021) and ConceptNet (Speer et al., 2017). The automatically generated knowledge may sometimes contain incorrect or irrelevant inferences. Here, we aim to understand how much COMET can contribute to our task. Hence we conduct oracle experiments where we choose the best possible inference from COMET and use it as an additional signal to integrate into the QA model.

We explore multiple ways of integrating this kind of external knowledge into a QA model for this task. In the first, we provide the model with some of the best possible relations as part of the input. This model is only required to generate the answer to the question. Next, we provide the model with the same input but also ask it to gener-

---

*Corresponding author: ylal@cs.stonybrook.edu

[1]We use COMET2020 for our experiments.

ate the best relation type along with the answer. In doing so, the model gives away the kind of knowledge it attends to in order to answer the question. Motivated by the gains resulting from integrating inferences from external sources, we finally build a model augmented by 5 COMET relations chosen according to the scores that COMET assigns it. This model, without access to any oracle information, shows improvement over previous models that do not cheat.

To analyze the relative merits of all these approaches, we first manually categorize the Why questions according to the types of knowledge that are needed to answer them. We find that most of the questions target Consequence, Goal seeking, Desire, and Reactionary types of knowledge. We use an Other category for the rest. We analyze the performance of different models across these knowledge categories. We compare the base and larger versions of T5 and its corresponding knowledge injected versions, where we use oracle relational inference from COMET.

We make the following observations: **(1)** performance improves dramatically when using the largest available model, **(2)** incorporating external knowledge shows substantial increases for smaller models but also provides significant increases even for a larger model, **(3)** external knowledge particularly helps on "implicit answer" questions where the answer is not explicitly stated in the story, **(4)** models seem to particularly lack the ability to utilize Consequence knowledge, and **(5)** a large model trained to jointly generate an answer as well as the type of relation needed to answer also performs well.

## 2 Related Work

### 2.1 Knowledge Bases

Knowledge bases (KBs) such as ConceptNet (Speer et al., 2017), WebChild (Tandon et al., 2017), Quasimodo (Romero et al., 2019) are examples of large knowledge bases constructed through semi-automated extraction over text, and contain world facts and informal relationships between common concepts that convey some prior knowledge. KBs that are compiled using crowdsourcing generally have a higher quality, e.g., ATOMIC (Sap et al., 2019) is an atlas of everyday commonsense knowledge and contains 880k triples about causes and effects of human activities and annotated by crowdsourced workers. ATOMIC is organized as if-then

relations and can be categorized based on causal relations (Sap et al., 2019; Hwang et al., 2021).

Petroni et al. (2019) show that rather than extracting knowledge from text or compiling using crowdsourcing, pretraining language models on text already endows them with certain types of factual knowledge that helps them do well on QA tasks. More recently, a popular approach is to fine-tune a language model on existing KBs, to generalize their knowledge and pays attention to the context, e.g., COMET (Bosselut et al., 2019; Hwang et al., 2021) generates context relevant commonsense knowledge. It is a fine-tuned language model over ATOMIC and ConceptNet KBs. Similarly, ParaCOMET (Gabriel et al., 2021) is a language model fine-tuned for discourse knowledge by fine-tuning over ROCStories, thus it generates relations consistent with an input narrative. We employ COMET as a source of commonsense knowledge in this paper.

### 2.2 Incorporating External Knowledge

Model outputs have been improved through commonsense injection during post-processing (Nag Chowdhury et al., 2018), using regularization at training time (Guan et al., 2020; Razniewski et al., 2021) and more recently by appending to the input (Lewis et al., 2020; Talmor et al., 2020) as recent models are judicious about the input context. Building upon the recent success of injecting commonsense in the input (Lewis et al., 2020; Talmor et al., 2020), our approach is to inject knowledge in the input by querying the knowledge source for task relevant knowledge. The semantic, contextual representation of the current commonsense sources helps alleviate past problems with searching for task specific knowledge in a static knowledge graph. There are two key challenges in using external sources. One is in figuring out what knowledge to use and the second lies in effectively integrating this into the end task.

Examples of recent research that inject triples into sentences in order to create domain-specific knowledge (Liu et al., 2020; Wang et al., 2020). Huang et al. (2019) incorporate commonsense knowledge directly into training data. Feng et al. (2020) leverage relations from ConceptNet using structured relational attention to perform multi-hop question answering. However, there is still uncertainty about the best way to represent external knowledge in order to solve commonsense reason-

| Model | Input Format | Output Format |
|---|---|---|
| T5 (NoCSK) | question: Why did Cam order a pizza? context: Cam ordered a pizza and took it home. He opened the box to take out a slice. Cam discovered that the store did not cut the pizza for him. He looked for his pizza cutter but did not find it. He had to use his chef knife to cut a slice. | answer: Cam was hungry. |
| T5 (OraCSK) | question: Why did Cam order a pizza? context: Cam ordered a pizza and took it home. He opened the box to take out a slice. Cam discovered that the store did not cut the pizza for him. He looked for his pizza cutter but did not find it. He had to use his chef knife to cut a slice. <info> they are hungry </info> | answer: Cam was hungry. |
| T5 UQA | why did cam order a pizza? \\n cam ordered a pizza and took it home. he opened the box to take out a slice. cam discovered that the store did not cut the pizza for him. he looked for his pizza cutter but did not find it. he had to use his chef knife to cut a slice. | cam was hungry. |
| T5 (OraCSK)UQA | why did cam order a pizza? \\n cam ordered a pizza and took it home. he opened the box to take out a slice. cam discovered that the store did not cut the pizza for him. he looked for his pizza cutter but did not find it. he had to use his chef knife to cut a slice. <info> they are hungry </info> | cam was hungry. |
| T5 (Ora5CSK)UQA | question: Why did Cam order a pizza? context: Cam ordered a pizza and took it home. He opened the box to take out a slice. Cam discovered that the store did not cut the pizza for him. He looked for his pizza cutter but did not find it. He had to use his chef knife to cut a slice. <info> relation: Causes  phrase: The pizza place is closed. </info>  <info> relation: oEffect  phrase: they eat it </info>  <info> relation: HasSubevent  phrase: the pizza is ready </info>  <info> relation: oEffect  phrase: gets hungry </info>  <info> relation: xWant  phrase: to eat it </info> | answer: Cam was hungry. |
| ExplainCOMET | question: Why did Cam order a pizza? context: Cam ordered a pizza and took it home. He opened the box to take out a slice. Cam discovered that the store did not cut the pizza for him. He looked for his pizza cutter but did not find it. He had to use his chef knife to cut a slice. <info> relation: Causes  phrase: The pizza place is closed. </info>  <info> relation: oEffect  phrase: they eat it </info>  <info> relation: HasSubevent  phrase: the pizza is ready </info>  <info> relation: oEffect  phrase: gets hungry </info>  <info> relation: xWant  phrase: to eat it </info> | relation: oEffect answer: Cam was hungry. |
| T5-11B (NoisyCSK) | question: Why did Cam order a pizza? context: Cam ordered a pizza and took it home. He opened the box to take out a slice. Cam discovered that the store did not cut the pizza for him. He looked for his pizza cutter but did not find it. He had to use his chef knife to cut a slice. <info> relation: Causes  phrase: The pizza place is closed. </info>  <info> relation: oEffect  phrase: they eat it </info>  <info> relation: HasSubevent  phrase: the pizza is ready </info>  <info> relation: oEffect  phrase: gets hungry </info>  <info> relation: xWant  phrase: to eat it </info> | answer: Cam was hungry. |

Figure 1: Example inputs and outputs for different models. The first four rows are used for both base and 11B sized models, while the last two are used only with 11B models.

ing problems (Zhang et al., 2020).

ERNIE (Zhang et al., 2019) is an enhanced language representation model trained using large-scale text corpora and knowledge graphs that shows significant improvements on various knowledge-driven tasks. Xiong et al. (2020) propose a weakly supervised pretraining objective, which explicitly forces the model to incorporate knowledge about real-world entities in order to perform entity-related question answering tasks. KGLM (Logan et al., 2019) is a neural language model with mechanisms for selecting and copying facts from a knowledge graph that are relevant to the context.

KagNet (Lin et al., 2019) grounds a question-answer pair in CommonsenseQA (Talmor et al., 2019) from the semantic space to the knowledge-based symbolic space as a schema graph, uses a KG-aware module to focus on it and scores answers with graph representations. Lv et al. (2020) propose a graph-based contextual representation learning module and a graph-based inference module to make better use of the graph information for commonsense question answering. DEKCOR retrieves information from ConceptNet and uses it to train an ALBERT model (Lan et al., 2020) for CommonsenseQA (Talmor et al., 2019) and Open-

BookQA (Mihaylov et al., 2018). Shwartz et al. (2020) generate and integrate background knowledge from pretrained LMs to present an unsupervised framework for multiple-choice commonsense tasks. Generated knowledge prompting elicits and integrates knowledge from language models using task-specific, human-written, few-shot demonstrations so as to improve performance on commonsense reasoning tasks (Liu et al., 2021).

## 3 Analyzing Commonsense in Models

Our goal is to analyze two sources of knowledge for reasoning about Why questions: (1) Large general-purpose LMs, effective models of distributional information which are shown to encode different kinds of knowledge, and (2) COMET, a commonsense specific knowledge (CSK) language model that contains many relations that are relevant to Why questions.

### 3.1 CSK in Pretrained LMs

Pretraining language models on text already endows them with certain types of knowledge (Petroni et al., 2019). So, we use three different versions of T5 to explore the capacity of large language models. In addition to the base T5 model

(T5-Base) with 220M parameters, we use the 11 Billion parameters version (T5-11B) as a large model. Both these versions are trained on a variety of tasks in addition to standard language modeling pretraining. Last, we also use the UnifiedQA checkpoint of the T5-11B model (T5-11B-UQA), which can be seen as a large model that is further fine-tuned for QA.

## 3.2 CSK in Finetuned LMs (COMET)

COMET is a transformer-based model that can generate commonsense inferences about events that it has learned from ATOMIC (Sap et al., 2019; Hwang et al., 2021) and ConceptNet (Speer et al., 2017). It provides commonsense knowledge across various dimensions for standalone events. It has been proven that such knowledge is helpful for various tasks. We try two methods of incorporating COMET knowledge into our models.

### 3.2.1 COMET Relations as Hints

One way to inject relevant knowledge into the model is to add the relevant COMET relations to the model's input. Our goal is to assess the potential for COMET's relations in answering Why questions. To this end, we first build an oracle that identifies the best relation generated by COMET for each question based on its semantic overlap with the answer for the question. For each sentence that was used to create a question in TellMe-Why, we obtain 3 relation phrases of different types from ATOMIC2020 (Hwang et al., 2021). We focus on relation types[2] about people (social interaction) and events (event-centered). We calculate the BertScore (Zhang* et al., 2020) between each relation and all gold answers for a question. The relation with the highest score is considered to be the best relation to help answer the question. We hypothesize that this is the kind of knowledge the model needs to answer the question correctly. The best relation is encapsulated inside an 'info' tag and concatenated to the end of model input. It is important to note that this is an oracle experiment since we use the gold answers in the test set to find the best relation for each question. We employ this oracle approach since our preliminary experiments showed that models have a hard time automatically determining the relevant type of knowledge when

---

[2]Full list of COMET relations: Causes, CausesDesire, DesireOf, Desires, HasFirstSubevent, HasLastSubevent, HasPrequisite, HasSubEvent, HinderedBy, MotivatedByGoal, oEffect, oReact, oWant, xEffect, xIntent, xNeed, xReact, xReason, xWant.

COMET inferences for all relations are included, resulting in decreased performance.

### 3.2.2 COMET Relations as Explanations

We explore another way to use the COMET knowledge injected into question answering models. First, for every question, each related COMET relation is ranked according to its BertScore with respect to all the gold answers for the question. We build a T5-11B model which takes the question, the context and the top 5 scoring COMET relations (in shuffled order) as its input. We call this model EXPLAINCOMET. It is trained to generate the answer to the question as well as the best relation type. Such a setup allows the model to automatically express the type of information it thinks is useful in answering the question. We compare this model with an analogous explicit injection method (T5-11B + top 5) where the input to the model is the same but the model is only required to produce the requisite answer.

### 3.2.3 COMET Relations as Noisy Hints

External knowledge sources are noisy and injecting them into a model as input increases their influence as well as risk for the task. Finally, we use COMET as a noisy source of information to train the T5 (NoisyCSK) model to better answer why questions in stories. For each question, we extract inferences from COMET along with its scores. The top 5 scoring relation types and phrases (in order) are used as supplementary information in the input of the model. We maintain the order of the relations to use the position bias (ranking), expecting the model to learn more from the higher ranked relations rather than the lower ranked ones. An example of its input-output behaviour can be seen in Figure 1.

## 4 Experiments

### 4.1 Dataset

TellMeWhy (Lal et al., 2021) is a dataset of 30k questions and free-form answers concerning why characters in short narratives perform the actions described. It is built upon the ROCStories corpus (Mostafazadeh et al., 2016). The questions are created by applying templates over events described in the narratives, and the answers are crowd-sourced from MTurk. Each question has 3 (possibly different) human answers. The dataset contains both explicit-answer questions (**Expl**; there is a possible

answer to the question in the narrative) and implicit-answer questions (**Impl**; the answer is not in the narrative, so external knowledge and/or reasoning is needed).

## 4.2 Implementation Details

First, we investigate T5[3] models (Raffel et al., 2020) that were designed to tackle a variety of text to text tasks, including free-form question answering. We follow the input format described in Appendix D.15 of Raffel et al. (2020) and illustrate an example in Figure 1. Next, we also analyse UnifiedQA models (Khashabi et al., 2020), variants of T5 that are tailored to question answering tasks. Each model is fine tuned in the same manner.

Finally, we analyse the T5 model in a multi-task setting. The model is given the question, the related story and its associated top 5 types and inferences from COMET. Essentially, we ask the model to generate an answer as well as an explanation (in the form of the COMET relation type and inference) for it.

## 4.3 Human Evaluation Metric

We use the human evaluation templates and MTurk settings provided by Lal et al. (2021) to collect judgments for the predicted answers of all the models. We asked the annotators whether the answer shown to them was valid. Each answer is evaluated by 3 annotators on a 5-point Likert scale (-2 to 2)[4]. We use the average Likert score over all answers as a metric for performance. The maximum score possible is 2, and the minimum is -2. In order to improve time and cost efficiency, we implement a caching mechanism to re-use previous annotator judgments for the same answer for a question in a particular story. For this purpose, we save all the human judgments for a (question, answer, story) triple. For all model predictions, we first check if a (question, answer, story) triple[5] is already present in the cache. If it is, we use the old judgments for it. If not, we gather validity annotations for it using human evaluation and add them to the cache for future use.

---

[3]Hereafter, unless specified otherwise, T5 refers to the 11 billion parameter version (T5-11B)

[4]Integer scores correspond to the labels: strongly disagree, disagree, neutral, agree, strongly agree

[5]All text is lowercased and answer is also stripped of punctuation.

| Model | Full | Expl | Impl |
|---|---|---|---|
| T5-BASE (NoCSK) | 0.19 | 0.56 | -0.56 |
| T5-BASE (NoCSK) UQA | 0.2 | 0.55 | -0.51 |
| T5 (NoCSK) | 0.91 | 1.11 | 0.51 |
| T5 (UQA) | **1.22** | **1.36** | **0.95** |
| Human | 1.35 | 1.39 | 1.28 |

Table 1: Effect of **model size**: Average likert score of human judgments of answers generated by models of different sizes. T5 denotes the T5 model, (UQA) denotes the UnifiedQA checkpoint for that size. (NoCSK) denotes that no commonsense knowledge was added to this model.

| Model | Full | Expl | Impl |
|---|---|---|---|
| T5-BASE (ORA1CSK) | 0.87 | 1.06 | 0.5 |
| T5-BASE (ORA1CSK) UQA | 0.75 | 0.88 | 0.51 |
| T5 (ORA1CSK) | **1.19** | **1.35** | **0.87** |
| T5 (ORA1CSK) UQA | 1.07 | 1.2 | 0.81 |
| Human | 1.35 | 1.39 | 1.28 |

Table 2: Effect of **explicit commonsense injection**: Average likert score of human judgments of answers generated by models with and without explicit commonsense knowledge injection. T5 denotes the T5 model, UQA denotes the UnifiedQA checkpoint for that size, and the ORACSK suffix denotes the model provided with the best relation during fine tuning.

## 5 Results

We presents human evaluation results that show the effects of model size (Table 1), explicit commonsense injection (Table 2), and implicit commonsense injection (Table 3).

**Effect of Model Size:** Table 1 shows the average Likert scores of models of different sizes. The base model performance is underwhelming, doing especially poorly on Impl. T5-base (UnifiedQA) is the best performing benchmark on the TellMe-Why dataset (Lal et al., 2021). We see that its performance is very similar to the T5-base model, showing only minor improvements on Impl. Next, we increase the model sizes to the 11 billion parameter T5 model. We find that the larger models show a notable performance improvement as compared to their base counterparts. In fact, T5-11B (UQA) comes close to human performance on Expl. While this large model comes close to human performance, there is still room for improvement for implicit answer questions.

**Effect of Knowledge Injection:** We add the

|  | Model | Full | Expl | Impl |
|---|---|---|---|---|
| **Explain** | EXPLAINCOMET | **1.31** | **1.4** | **1.13** |
| **Answer** | T5 (ORA5CSK) | 1.28 | 1.39 | 1.11 |
|  | Human | 1.35 | 1.39 | 1.28 |

Table 3: Effect of **explaining model answers**: Average likert score of human judgments of answers generated by EXPLAINCOMET model (built over T5) and a large model given the top 5 relations (ORA5CSK) in shuffled order.

| Model | Full | Expl | Impl |
|---|---|---|---|
| T5 (NOCSK) | 0.91 | 1.11 | 0.51 |
| T5 (NOISYCSK) | **1.09** | **1.25** | **0.76** |
| T5 (ORA1CSK) | 1.19 | 1.35 | 0.87 |
| T5 (ORA5CSK) | 1.28 | 1.39 | 1.11 |
| Human | 1.35 | 1.39 | 1.28 |

Table 4: Effect of **using top 5 relations**: Average likert score of human judgments of answers generated by the NOISYCSK model and a large model given the top 5 relations in shuffled order (ORA5CSK). The numbers in bold represent the performance of a model without any oracle information.

best scoring relation (*top1*) to the input of the models to provide them extra information to answer the question. This *top1* relation is the relation that has the highest BertScore with respect to all the gold answers for the question. Table 2 shows the average Likert scores of models which had access to external commonsense. Compared to Table 1, we find that adding this information helps the base models answer questions a lot better. It also improves the performance of T5-11B, bringing it close to human performance on Expl. However, it *hurts* T5-11B (UQA) performance significantly. We hypothesize that, since this model architecture is pretrained for question answering, it is possible that providing any information other than context distracts it from its ability to do the task. Wu et al. (2021) previously have shown an instability in performance when scaling up T5 (UnifiedQA) models. Our findings are in line with that. Overall, even with the best possible information from an external source, there is a significant gap with human performance on the implicit answer questions.

**Effect of Explaining Model Answers:** Having shown that models benefit from commonsense knowledge, we proceed to explore a way to integrate this kind of information into the model implicitly. Table 3 shows that ExplainCOMET achieves overall performance close to humans on this task. In fact, it does slightly better on Expl. But there is still a clear gap on the implicit answer questions. In addition to that, it is able to generate the correct best relation type 44.27% of the time.

To demonstrate the advantage of implicit injection, we compare this joint training method with an analogous explicit injection method. We see that T5-11B + top 5, despite having explicit access to the top 5 scoring relations extracted from COMET, scores a little lower than ExplainCOMET model for all types of questions. It should be noted that both these models cheat to a lesser extent as they do not have direct access to the best relation in-

formation. Instead, during training, they learn to attend to relevant knowledge for answer generation. This requires some understanding of the type of knowledge needed to answer a question.

**Effect of using Noisy Hints:** Having shown an upper bound for using COMET relations as a source of commonsense knowledge for why question answering, we proceed to build a model that tries to leverage them. We use the top 5 relations from COMET (as scored by COMET itself) and add that information into the input encapsulated in '<info>' tags. Table 4 shows that this model's performance lies squarely between a model trained without external commonsense and a model trained with ideal commonsense knowledge.

## 6 Analysis

To better understand the strengths and weaknesses of these models, we defined an ontology for the types of knowledge that are required to answer TellMeWhy questions. We identified five categories of answers, and then labeled the CATERs subset of TellMeWhy, for which the gold answers already have judgments. The categories are: **(1)** Goal-seeking: an agent performed an action because it was an intermediate step to a larger goal (22.6% of questions), **(2)** Reactionary: an agent performed an action as a reaction/followup to another event (23.1% of questions), **(3)** Desire: an agent performed an action to accomplish an inherent goal (11.5%), **(4)** Consequence: an event (a tangible action was not performed) happened as a consequence of another event (35.6%), **(5)** Other: types of knowledge that do not fall into the categories above (7.4%).

Since there is a bigger gap with human performance on the implicit answer questions (Impl), we analyse them to further understand the gaps in the

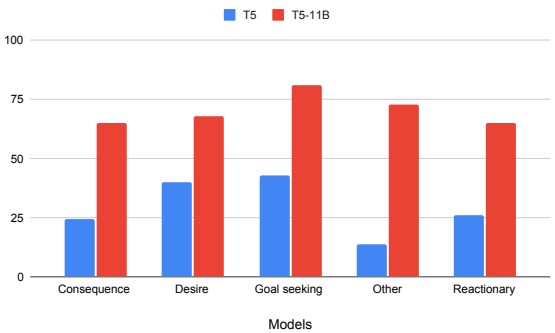

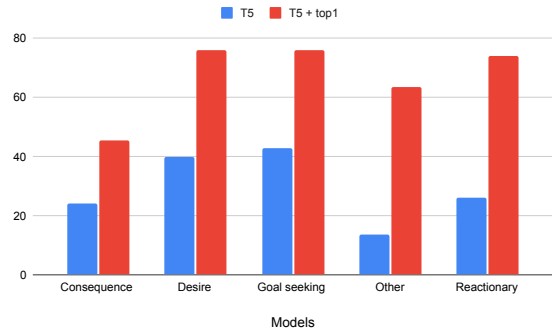

Figure 2: Accuracy of the base and larger models on different ontological types of questions.

Figure 4: Effect of adding external commonsense knowledge to T5-base.

models' understanding and to identify possible areas for improvement. We present the distribution of performances of the different models on each other categories. To further quantify the differences across models, we first compute a failure probability for each category i.e., the probability of a failed question to belong to a given category. We compute this by dividing the number of incorrectly answered questions of that knowledge type by the total number of questions it gets wrong. We measure the differences in these failure probability distributions across two models using the Jensen-Shannon Divergence (JSD).

### 6.1 What kind of information do humans use that are inaccessible to models?

Figure 2 shows accuracy for the different types of questions. The base model is unable to reason adequately about the 'Consequence' and 'Other' types of knowledge. However, as the model size increases, it is apparently able to capture a variety

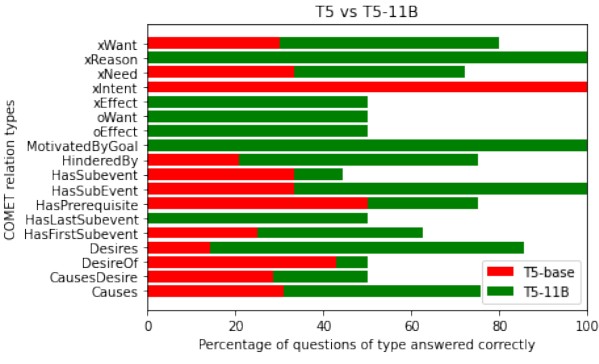

Figure 3: Effectiveness of T5 models at capturing different types of relational knowledge. Both models correctly answer the same number of questions for some relation types (indicated by the presence of only one color).

of types of knowledge and it also demonstrates the gaps in their understanding as compared to humans. Understanding all consequences of an event is difficult, so it is plausible that it is the hardest category for models to learn.

The failure probability distributions for UQA and UQA-11B have a moderate divergence. The JSD of the distributions across categories is only 0.14, suggesting that there is only a slight difference in the kinds of knowledge both models are unable to capture.

### 6.2 What kinds of COMET relations are already accessible to larger models?

We can also categorize the questions in terms of the COMET relation type that best helps to answer the question. Using this wee can analyze what kinds of knowledge seem to already be encoded in the larger model that allows it to answer questions better than the smaller model. Comparing this with Figure 3 shows that the larger model seems to capture many types of COMET relations. In fact, increasing the size of the model helps it accurately answer all the questions for some relation types (HasPrerequisite, HasLastSubEvent, HasSubEvent). However, it does not help for information related to effects (xEffect, oEffect), amongst others. It is clear that there is a lot of ground to be covered for most relation types.

### 6.3 What kind of questions does external knowledge help with?

There is a large potential for improvement if we can effectively integrate external knowledge into models as shown in Table 2. In Figure 4, we see that adding external knowledge helps the base model improve the most on Consequence and Other, although these also had the most overall room for

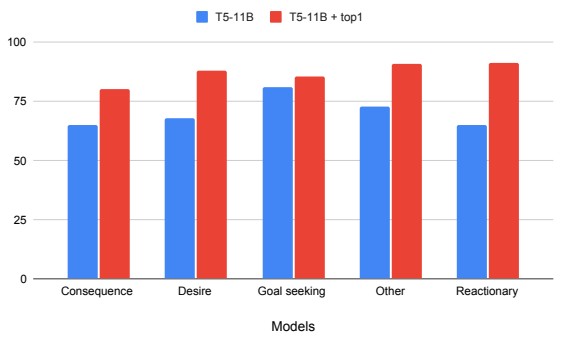

Figure 5: Effect of adding external commonsense knowledge to T5-11B. Both models answer the same number of questions requiring Other kinds of knowledge.

improvement to begin with. In terms of errors, the JSD between the failure probability distributions of the base and larger models is 0.24, which suggests that there is a larger divergence in model's abilities to use knowledge of different categories. Figure 2 demonstrates that larger models are able to capture these kinds of information better than the base model (even with the external knowledge added to it), showing that they already contain the some of the external commonsense knowledge.

For the UnifiedQA model, we observe that adding external knowledge to the larger model actually hurts performance. We hypothesize that since UnifiedQA is trained on question answering tasks using only the context and question, it is plausible that this extra information confuses the model from learning the right micro-patterns. External commonsense helps learn more 'Reactionary' and 'Other' knowledge but confuses the model about all other kinds of knowledge.

## 7 Conclusion and Future Work

Answering Why questions requires access to some forms of commonsense knowledge. This work analyzed how much of this knowledge is already accessible in large models, what parts of it can be tapped from the COMET commonsense relations. As we would expect, large models seem to contain a larger portion of this knowledge compared to the base size model. But we also find that question relevant COMET relations have the potential to substantially improve performance even for a large model. The knowledge category analyses shed further light on what kinds of knowledge are helpful.

Our empirical study indicates that these commonsense sources usually contain the required

knowledge, but it is not easy to tell apart the task relevant knowledge just by using the scores of those sources. Future work needs to develop better ways of automatically locating relevant relations in order to realize the potential. We also show that a simple approach for commonsense injection has to deal judiciously with the noise in the commonsense source. In the absence of any additional supervision signal, this noise limits the learning of the model hence we need more advanced methods that can deal with the inevitable noise in commonsense sources.

We demonstrate better ways to train to deal with some noise when provided the extra supervision signals of the expected explanation (our Explain-COMET model). Such signals are not present at test time and thus our current models provide an upperbound. We show that it is possible in the near-future to close the gap w.r.t. human performance on simple story tasks, but this would require new techniques that can jointly learn to distinguish the noise from existing sources of commonsense while leveraging their redundant signals.

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
