# OpenReview forum: "Analyzing the Contribution of Commonsense Knowledge Sources for Why-Question Answering"
_aclweb.org/ACL/2022/Workshop/CSRR — ACL 2022 Workshop CSRR_

### Official Review · Reviewer_5cXn · 2022-03-19

**Rating:** 7
**Confidence:** 4

**Review:**

Summary
- This paper considers the QA task of Why questions, which requires commonsense knowledge. The paper studies (1) what aspects of this knowledge is accessible to pretrained language models, and (2) what aspects can be made accessible via external commonsense resources such as COMET. The authors show that the QA performance can be improved by (a) increasing the LM size, (b) injecting knowledge from COMET as part of the task input, and (c) teaching the model to generate COMET relation type as output besides the answer. The paper conducts extensive experiments, and analyzes what types of commonsense knowledge pretrained LMs already have, and what can be augmented via COMET.

Reasons to Accept:
- The paper asks interesting and important questions (what knowledge pretrained LMs already have, what can be gained from external knowledge, what knowledge remains inaccessible), and answers them with sufficient experiments.
- The paper is clear and well-written.

Weakness and questions:
- Do you have an intuition why does the larger model (T5 11B) captures more COMET relations than smaller model (T5 base)? Was T5 11B trained on more data, or simply the model parameter count is larger and has more capacity to cover more relations? Why T5 11B is good at capturing particular relations (e.g. xReason) but not others (e.g. HasSubEvent)?

Typos/grammar:
- L234: "both these" -> "both of these"
- L503: "wee" -> "we"

---

### Official Review · Reviewer_w3mr · 2022-03-24
**Using COMET for TellMeWhy**

**Rating:** 7
**Confidence:** 5

**Review:**

In this paper, the authors study Why questions in the TellMeWhy dataset using COMET as a source of relevant commonsense relations. They analyze the relative improvements over a base T5 model when (a) increasing the model size, (b) injecting knowledge from COMET as part of the task input, and (c) asking the model to generate COMET relation type as an explanation in addition to its answer. Their results show that the larger model, as expected, yields substantial improvements over the base. Interestingly, they find that the question specific COMET relations can provide substantial improvements for both base and large models, with additional possible gains when asking the model to also generate COMET relation type.  So, they augment a large model with noisy hints from COMET and find that this improves performance on the  task.

Overall this is decent work. I don't have any particular criticism, however, I have to bring this up that at this point using commonsense knowledge from COMET ATOMIC at the realm of reasoning in narratives is not particularly novel because several works have done this. such as abductive reasoning work by Bhagavatula et al. (2019) for abductive reasoning, and self talk by Shwartz et al. (2020)

There are couple of follow up questions
1) Since Tell me why is focused on narratives why not use ParaCOMET which is specifically designed for narratives and is discourse aware
2) Did you also look into concept centric commonsense?
3) Your related work on  Incorporating External Knowledge is great but there needs to be more discussion on prior work which used COMET such as
i)  Ammanabrolu et al. (2020) for story generation,
          Automated storytelling via causal, commonsense plot ordering
ii)  Majumder et al. (2020) for dialog generation
         Like hiking? you probably enjoy nature: Persona-grounded dialog with commonsense expansions.
iii) Chakrabarty et al. (2020a; 2020b; 2022)
         Rˆ3: Reverse, Retrieve, and Rank for Sarcasm Generation with Commonsense Knowledge
         Generating similes effortlessly like a Pro: A Style Transfer Approach for Simile Generation
         It’s not Rocket Science: Interpreting Figurative Language in Narratives

---

### Official Review · Reviewer_xGYV · 2022-03-25
**An interesting study with minor weaknesses**

**Rating:** 7
**Confidence:** 5

**Review:**

Summary:

The authors present a very interesting analysis of the extent to which common sense knowledge is embedded/used with large language models (LLMs) - on a narrative task. Specifically, authors aim to address the following key important questions: (a) aspects of common sense knowledge accessible to LLMs; (b) aspects of common sense that could be leveraged from signals obtained through external knowledge graph (KG) sources like COMET.  Experimental findings show that increasing size of the LLM model helps in capturing common sense better; incorporating external KG helps in improving model performance in answering questions that are not directly available in the narratives.


Strengths:

The paper is well-organized, well-written and easy to follow.

I thoroughly enjoyed reading this work and I am sure that this work would be a great fit for this workshop audience.

Extensive experiments have been performed to draw several interesting insights on the problem setting.


Weaknesses:

I did not find any major weaknesses with this work.

Can these findings be generalized to other datasets/domains? It would have been really great to have one more dataset (preferably on a similar but different task), to better quantify the conclusions made in the paper.

In section 6, I didn’t exactly understand the differences between goal-seeking and desire categories. Are you planning to provide supplemental material with this effort? More fine-grained categories would make this analysis even stronger.

I struggle to identify the differences between the experiment setup described in section 3.2.1 and 3.2.2. It would help to make this more clear, perhaps an example would help.

---

### Decision · Program_Chairs · 2022-03-28

Accept